# Dengue viruses cleave STING in humans but not in nonhuman primates, their presumed natural reservoir

Alex C Stabell[1], Nicholas R Meyerson[1], Rebekah C Gullberg[2], Alison R Gilchrist[1], Kristofor J Webb[1], William M Old[1], Rushika Perera[2], Sara L Sawyer[1]*

[1]Department of Molecular, Cellular and Developmental Biology, University of Colorado Boulder, Boulder, United States; [2]Arthropod-borne and Infectious Diseases Laboratory, Department of Microbiology, Immunology and Pathology, Colorado State University, Fort Collins, United States

**Abstract** Human dengue viruses emerged from primate reservoirs, yet paradoxically dengue does not reach high titers in primate models. This presents a unique opportunity to examine the genetics of spillover versus reservoir hosts. The dengue virus 2 (DENV2) - encoded protease cleaves human STING, reducing type I interferon production and boosting viral titers in humans. We find that both human and sylvatic (reservoir) dengue viruses universally cleave human STING, but not the STING of primates implicated as reservoir species. The special ability of dengue to cleave STING is thus specific to humans and a few closely related ape species. Conversion of residues 78/79 to the human-encoded 'RG' renders all primate (and mouse) STINGs sensitive to viral cleavage. Dengue viruses may have evolved to increase viral titers in the dense and vast human population, while maintaining decreased titers and pathogenicity in the more rare animals that serve as their sustaining reservoir in nature.
DOI: https://doi.org/10.7554/eLife.31919.001

*For correspondence: ssawyer@colorado.edu

## Introduction

Dengue viruses cause clinical disease in approximately 100 million individuals each year and are found in over 100 countries (*Bhatt et al., 2013*). Yet, to date no vaccine exists that conveys cross-protection against all human dengue viruses (*Scherwitzl et al., 2017*). Dengue viruses are positive sense RNA viruses in the family *Flaviviridae*, and are related to yellow fever virus, Zika virus, and West Nile virus (*Best, 2016*). These viruses are primarily transmitted between humans in highly populated areas by *Aedes aegypti* and *Aedes albopictus* mosquitoes, in what are referred to as human (or 'urban') transmission cycles (*Diamond and Pierson, 2015*; *Hanley et al., 2013*; *Vasilakis et al., 2011*). Sylvatic (i.e. forest) dengue virus transmission cycles, which are separate from the human transmission cycles, exist in Asia and Africa and involve nonhuman primates and forest-dwelling *Aedes* mosquitos (*Vasilakis et al., 2011*; *Wang et al., 2000*; *Rico-Hesse, 1990*). While the exact nonhuman primate species that serve as the sustaining natural reservoirs for sylvatic dengue viruses are unknown, the global distribution of both dengue viruses and their transmitting mosquitoes could be consistent with a significant number of primate species being involved (*Figure 1—figure supplement 1*) (*Hanley et al., 2013*; *Vasilakis et al., 2011*). Primarily, dengue viruses have been associated with monkeys (rather than apes) found in Africa and Asia (*Figure 1*). Human dengue viruses cluster into four phylogenetically distinct clades referred to as DENV1, 2, 3, and 4 (*Vasilakis and Weaver, 2008*). These clades have sylvatic dengue virus isolates at their bases, supporting zoonotic origins of the four dengue viruses that now circulate in humans (*Wang et al., 2000*; *Pyke et al., 2016*;

**eLife digest** Dengue viruses are found in over 100 countries and cause the tropical disease known as dengue fever. Dengue viruses affect around 100 million people per year and can – in severe cases – lead to death. Unlike many other deadly diseases, there is currently no vaccine that completely prevents dengue fever.

It is thought that dengue viruses that circulate in human populations were derived from monkey versions of that same virus. However, research suggests that both human and primate variations of dengue viruses appear to multiply much better in humans than in other species. Scientists believe that this is because some animals, including primates, have defense mechanisms that are ineffective in humans.

To explore this idea, Stabell et al. looked at a protein called STING in humans and in three different primates: the chimpanzee, the rhesus macaque, and the common marmoset. STING plays an important role in the immune system and helps to fight infections caused by viruses and other microbes.

During replication – the process by which a virus spreads through an organism's cells – the dengue virus cuts and inactivates the human STING protein, and so helps the virus spread. Stabell et al. discovered that in most primates, dengue viruses cannot inactivate STING. This was found to be reliant on a small region in the STING protein that differed between humans and primates. This small difference may, in part, explain why dengue viruses replicate better in humans than other primates.

Stabell et al. then searched for other animals whose STING protein would be susceptible to dengue virus inactivation. Using a database with genetic information of over 5,000 mammals, Stabell et al. identified STING proteins of three types of apes and three types of rodents that could also be deactivated by dengue viruses.

To develop a vaccine or antiviral drug scientists generally need to study the disease in living animals. Since dengue viruses replicate more successfully in humans than they do in other animal models, it makes it more challenging to find an effective treatment. The results from Stabell et al. may help to identify animals that could be strong candidates for future research into dengue viruses, potentially paving the way for further therapeutic development.

DOI: https://doi.org/10.7554/eLife.31919.002

Weaver and Vasilakis, 2009). Human dengue viruses have now become uncoupled from the sylvatic reservoir and require only humans and mosquitoes to be sustained (*Mayer et al., 2017*).

In side-by-side experiments, sylvatic and human dengue viruses replicate similarly in human cells (*Vasilakis et al., 2007*; *Vasilakis et al., 2008*). These results have been interpreted to mean that there is little or no adaptive barrier for the emergence of sylvatic dengue viruses into human populations, and the view that dengue viruses are generalists capable of infecting a wide range of primate species including humans. Thus, a paradox exists in understanding why human dengue viruses are so difficult to model in nonhuman primates. Chimpanzees (*Pan troglodytes*) (*Scherer et al., 1978*), rhesus macaques (*Macaca mulatta*) (*Halstead et al., 1973*; *Hickey et al., 2013*), marmosets (multiple *Callithrix* species) (*Moi et al., 2013*; *Ferreira et al., 2014*), and other nonhuman primate species (*Althouse et al., 2014*) have been explored as possible primate models for studying dengue virus pathogenesis and for vaccine challenge. In general, it has been observed that dengue does not replicate to high titers in these models, and little or no overt disease pathology is observed (*Cassetti et al., 2010*; *Zompi and Harris, 2012*). If human and sylvatic viruses are the same in their properties, we speculated that there must instead be something special about the replication of these viruses in the human host.

STING is a multi-pass transmembrane protein found in the endoplasmic reticulum, and functions as a critical component in the innate immune sensing pathway for intracellular pathogens (*Ishikawa and Barber, 2008*; *Zhong et al., 2008*; *Ishikawa et al., 2009*; *Jin et al., 2008*; *Sun et al., 2009*; *Burdette and Vance, 2013*). Although originally described as part of the response to cytosolic DNA sensing (*Zhang et al., 2011*), STING is also activated upon RNA virus infection (*Holm et al., 2016*). Underscoring this, several RNA viruses encode proteins that antagonize or

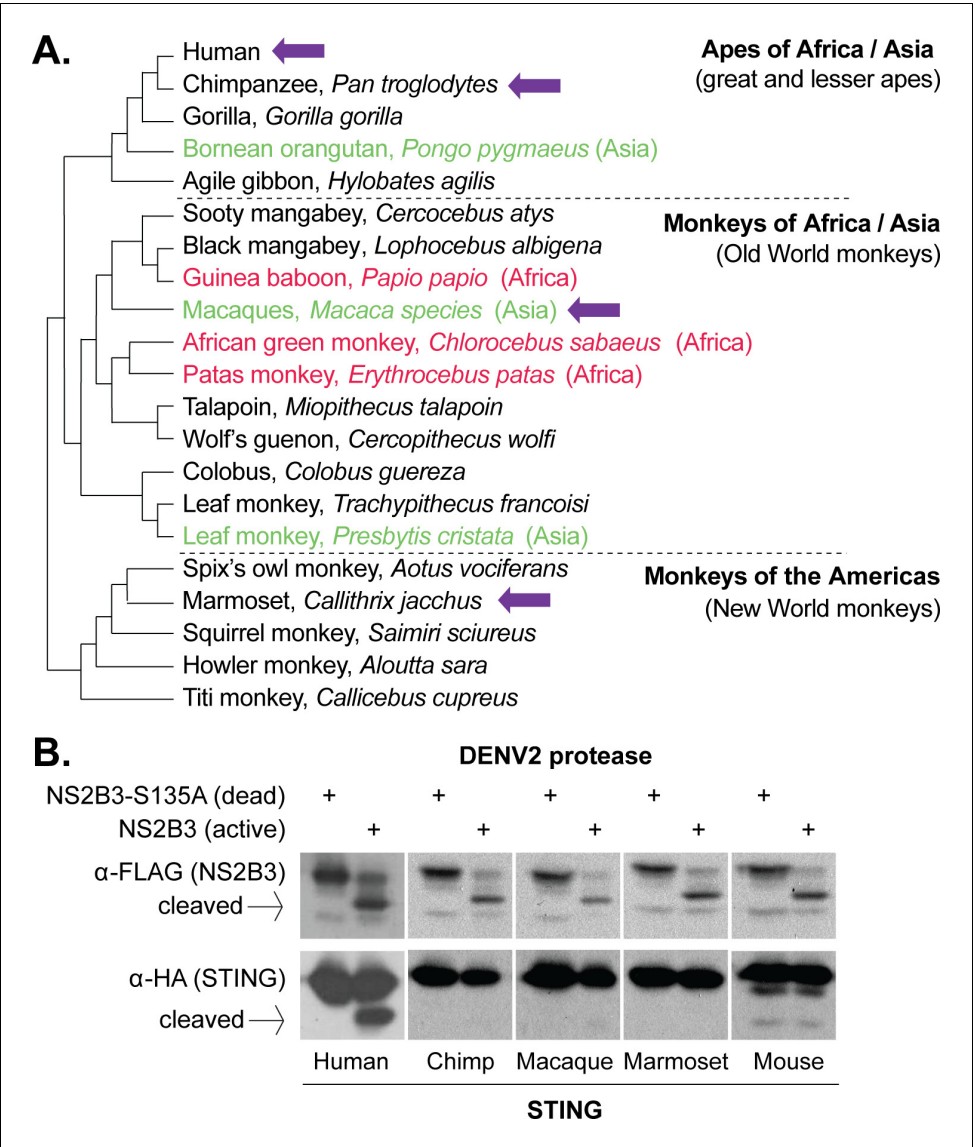

**Figure 1.** Dengue virus (DENV2) can cleave human but not nonhuman primate STING. (**A**) A phylogeny of select primate species, showing the three main simian clades: apes, Old World monkeys, and New World monkeys (**Perelman et al., 2011**). The primate species from which STING is tested in this study are shown with purple arrows. Possible primate reservoir hosts for sylvatic dengue viruses, based on virus isolation from sentinel monkeys, or antibody detection, are shown in red (Africa) and green (Asia). The current evidence for these primate reservoir hosts is reviewed in the discussion section. (**B**) 293T cells were cotransfected with plasmids encoding STING-HA, and the NS2B3-Flag protease complex with or without the S135 inactivating mutation. Whole cell lysate isolated 24 hr post transfection was run on a protein gel and immunoblotted with anti-Flag or anti-HA antibodies. The encoded NS2B-NS3-Flag polyprotein auto-processes into the NS2B3 protease complex if the protease is active, as seen in the anti-Flag blot where in some samples the NS3-Flag protein has been liberated through cleavage. We sometimes see lower bands underneath the full-length mouse STING, but conclude that they are endogenous degradation products since they are equal in intensity in the presence of the active or dead protease.

DOI: https://doi.org/10.7554/eLife.31919.003

The following figure supplement is available for figure 1:

**Figure supplement 1.** Many primate species reside in areas where dengue viruses are endemic in humans.
DOI: https://doi.org/10.7554/eLife.31919.004

degrade STING (*Sun et al., 2012*; *Nitta et al., 2013*; *Ding et al., 2013*; *Aguirre et al., 2012*; *Yu et al., 2012*). For instance, the NS2B3 protease of one human dengue virus, DENV2, has been shown to target human STING for cleavage (*Aguirre et al., 2012*; *Yu et al., 2012*). Through the cleavage of STING, DENV2 renders the host unable to induce the phosphorylation of Interferon Regulatory Factor 3 (IRF3), therefore decreasing production of type I interferon and increasing viral titers (*Green et al., 2014*). Mouse STING is resistant to cleavage by the DENV2 protease (*Aguirre et al., 2012*; *Yu et al., 2012*). This at least partially explains why mice mount an effective immune response against dengue viruses, protecting them against infection and compromising their utility as model organisms (*Cassetti et al., 2010*; *Zompi and Harris, 2012*; *Ashour et al., 2010*). Dengue viruses are known to mute the host interferon response in other ways as well, with the other predominant mechanism being the degradation of STAT2 (*Ashour et al., 2009*; *Jones et al., 2005*; *Mazzon et al., 2009*; *Best, 2017*; *Morrison et al., 2012*).

In this study, we show that the NS2B3 proteases of human (DENV1-4) and sylvatic dengue viruses universally cleave human STING. However, none of these proteases can cleave the STING proteins of chimpanzees, macaques, or marmosets, three primate species that have been pursued as model organisms. We show that an 'RG' motif at positions 78/79 of STING is critical for susceptibility to cleavage, and conversion of these residues to 'RG' renders all nonhuman primate STING proteins tested, as well as mouse STING, sensitive to dengue virus proteases. Out of the entire Genbank database, along with our sequencing of STING from 16 additional primate species, we identify only a small number of apes (gorillas, orangutans, and gibbons), and three small rodent species (chinchillas, naked mole rats, and desert woodrats) as encoding a functional dengue virus cleavage determinant in STING. This may, in part, explain why modeling dengue virus in animal models has been so difficult.

## Results

### The protease of human dengue virus, DENV2, cleaves only human STING

To begin, we cloned STING from chimpanzee (*Pan troglodytes*, Genbank XM_016953921), rhesus macaque (*Macaca mulatta*, Genbank MF622060), and the common marmoset (*Callithrix jacchus*, Genbank MF622061). These species have been explored as animal models of dengue infection, and also represent the three major clades of simian primates: apes (represented by chimpanzee), Old World monkeys (represented by macaque), and New World monkeys (represented by marmoset; *Figure 1A*). Most suspected dengue virus reservoir hosts belong to the Old World monkey clade (red and green type in *Figure 1A*). On the other hand, New World monkeys (such as marmosets), which reside exclusively in the Americas, have presumably never been exposed to sylvatic dengue viruses since sylvatic cycles do not exist in the New World. We also included human (Genbank MF622062) and mouse (*Mus musculus*, Genbank MF622063) STING in our studies as positive and negative controls, since it was previously shown that human but not mouse STING is sensitive to DENV2 NS2B3 cleavage (*Aguirre et al., 2012*; *Yu et al., 2012*).

The dengue virus NS2B3 protease complex is composed of the viral non-structural proteins NS2B and NS3 (*Preugschat et al., 1990*; *Zhang et al., 1992*; *Falgout et al., 1991*). In the dengue virus genome, the NS2B and NS3 genes sit adjacent and are cotranslated as part of a single long viral polyprotein (*Perera and Kuhn, 2008*; *Chambers et al., 1990*). When the NS2B - NS3 region is expressed from a plasmid, the region is translated into a small polyprotein that then auto-cleaves itself to become the functional protease complex (*Yusof et al., 2000*; *Bera et al., 2007*). We used a plasmid expressing the NS2B-NS3 region, including a 3x Flag tag at the C-terminus of NS3, from the New Guinea C isolate of DENV2 (see methods). As a control, a mutation was created at the active-site serine, changing it to an alanine (S135A), which renders the protease inactive (*Rodriguez-Madoz et al., 2010*). We then used a previously established cotransfection assay (*Aguirre et al., 2012*; *Yu et al., 2012*) to determine if the dengue virus protease could cleave primate STING orthologs. Plasmids encoding primate or mouse STING, and either active or S135A (dead) NS2B3 dengue proteases, were cotransfected into 293T cells. STING cleavage was assessed 24 hr later by western blot. The inactivity of the S135A protease can be seen in the anti-Flag blot, where the NS2B-NS3 polyprotein does not self-cleave when this mutation is present (*Figure 1B*). We see only a fraction of

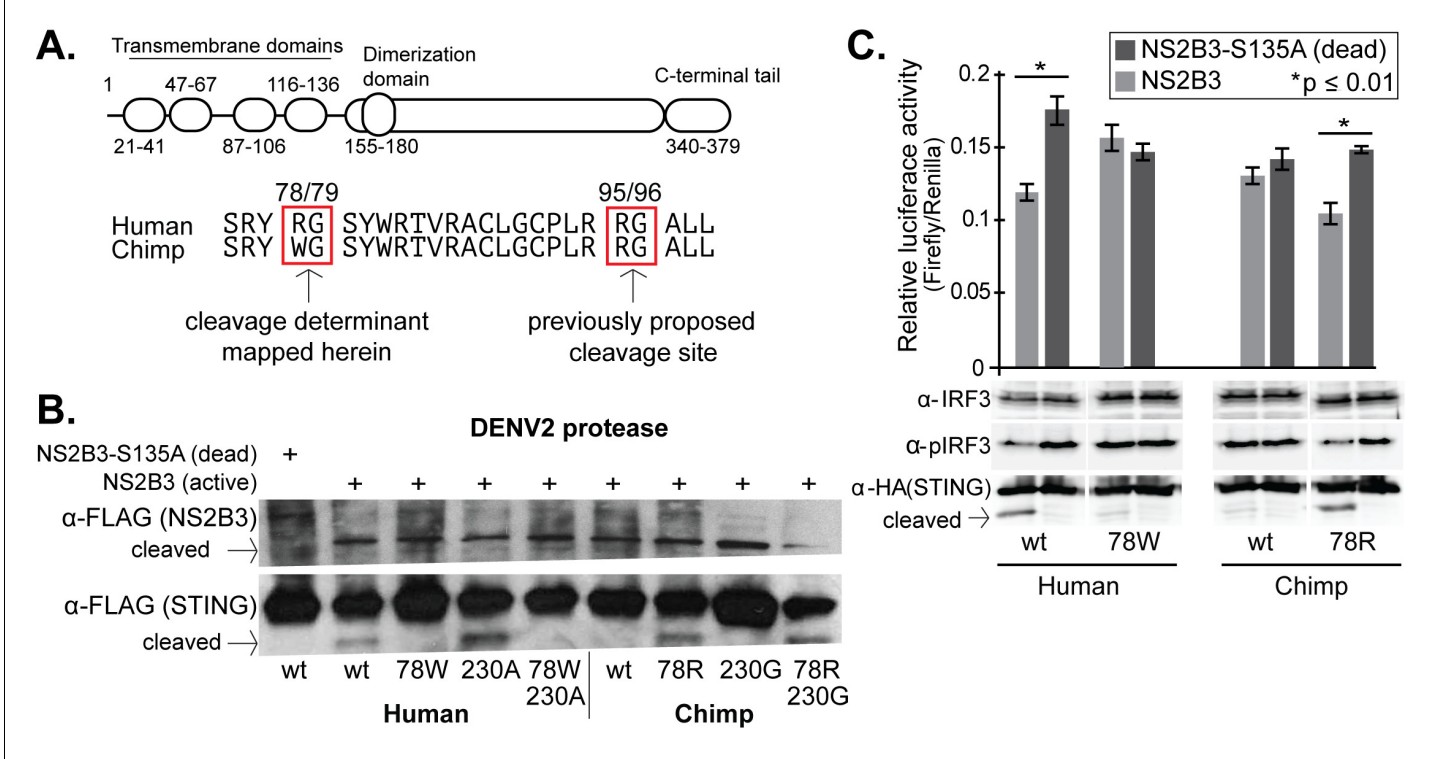

**Figure 2.** STING residue 78 determines susceptibility to NS2B3 cleavage in human versus chimpanzee STING comparisons. (**A**) A domain diagram of human STING is shown, as defined in (**Wu et al., 2014**). An alignment of human and chimpanzee STING in the region of the newly identified cleavage determinant (78/79) and the one previously determined (95/96) (**Aguirre et al., 2012**; **Yu et al., 2012**). (**B**) Site-directed mutagenesis was performed on either human or chimpanzee STING at position 78, substituting the residue at this position in human (R) with that in chimpanzee (W) and vice versa. Plasmids encoding the NSB3 protease complex and STING were cotransfected into 293T cells, and 48 hr later lysates were collected and analyzed by anti-FLAG western blot. In this experiment, both the protease and STING are tagged with FLAG. Data presented are representative of at least two experiments. (**C**) (bottom) 293T cells were transfected with plasmids expressing the DENV2 NS2B3 protease and wildtype (wt) or mutated (78W or 78R) STING. IRF3 and phosphorylated IRF3 (pIRF3) were detected by western blot in lysates harvested 48 hr later. (top) The identical experiment, but performed in biological triplicate and with the addition of plasmids encoding a firefly luciferase gene driven by the interferon beta (IFNb) promoter, and a renilla luciferase gene driven by a CMV promoter. The relative luciferase activity (Y-axis) was calculated by normalizing the luciferase signal to the renilla signal in each replicate. A Welch's T-test was used to compare the levels of luciferase produced in the presence of active versus dead protease. Data is representative of at least two experiments.

DOI: https://doi.org/10.7554/eLife.31919.005

The following figure supplements are available for figure 2:

**Figure supplement 1.** Generation of STING knockout cells using CRISPR-Cas9, and stable re-complementation of these lines.
DOI: https://doi.org/10.7554/eLife.31919.006

**Figure supplement 2.** Wildtype and mutated human STING both colocalize with the ER-resident protein BiP.
DOI: https://doi.org/10.7554/eLife.31919.007

the human STING being cleaved, but this is consistent with previous publications and is presumably exacerbated by the overexpression of STING achieved in transfection experiments (**Aguirre et al., 2012**; **Yu et al., 2012**). Unexpectedly, none of the nonhuman primate STINGs tested were susceptible to cleavage (**Figure 1B**). Remarkably, the DENV2 protease could not even cleave chimpanzee STING, which differs from human STING at only three amino acid positions.

## Mapping the dengue virus cleavage determinants in STING

The dengue virus cleavage site in STING was previously mapped to between the 95th and 96th residues (**Aguirre et al., 2012**; **Yu et al., 2012**). Some uncertainty existed, though, because in the previous studies it was noted that the human residues around 95/96 were not sufficient to convey cleavage susceptibility to mouse STING. Indeed, human and chimpanzee STING proteins have the exact same amino acid sequence at these positions (**Figure 2A**). Human and chimpanzee STING

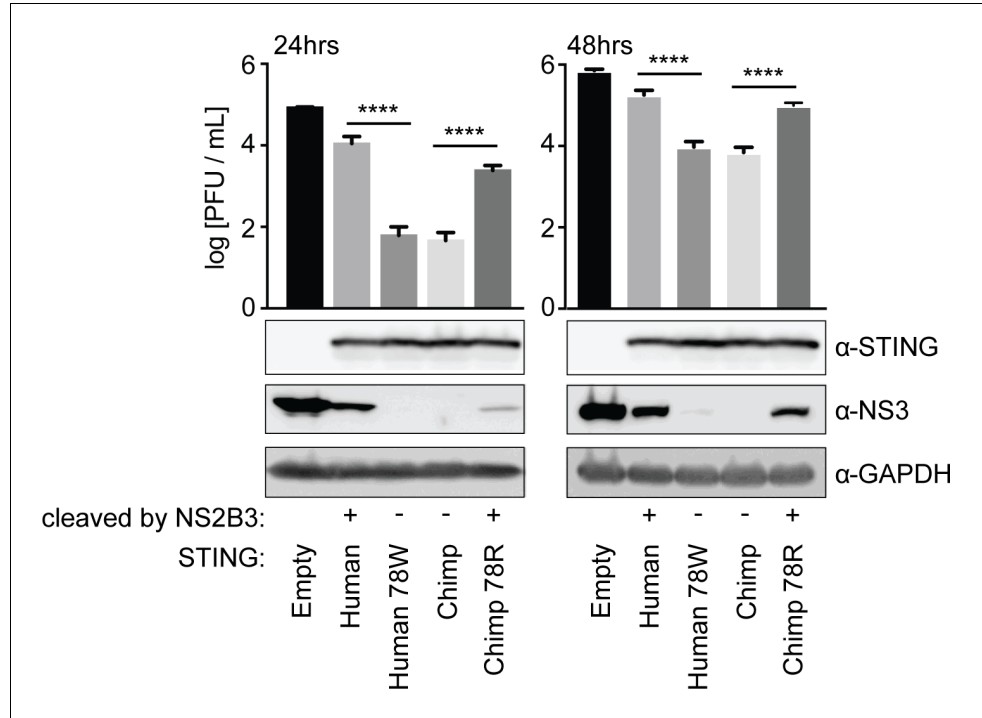

**Figure 3.** Cleavage of STING at position 78/79 promotes virus replication. The endogenous copies of STING in A549 cells were knocked out using the Cas9 nuclease (see *Figure 2—figure supplement 1*). These cells were re-complemented by retroviral transduction with no gene (pLPCX-empty), wildtype human STING, cleavage-resistant human STING (human 78W), wildtype chimpanzee STING, or cleavage-susceptible chimpanzee STING (chimp 78R). These cell lines were infected at MOI of 0.3 with dengue virus 2 (DENV2 16681). After 24 and 48 hr the virus supernatant was removed and titrated on BHK21 cells. At the same time, cells were collected in RIPA buffer, lysed, and run on a gel for western blotting using antibodies against STING, dengue virus NS3, and GAPDH (loading control). A Tukey's multiple comparisons test indicated significant differences in infectious virus in the presence of each mutant STING compared to wildtype STING, as shown (****=p < 0.0001), after significant one-way ANOVA. Data are representative of at least two independent experiments.

DOI: https://doi.org/10.7554/eLife.31919.008

The following figure supplement is available for figure 3:

**Figure supplement 1.** STING is cleaved during dengue virus infection.

DOI: https://doi.org/10.7554/eLife.31919.009

differ at only three amino acid positions, residues 78, 230, and 232. We found that mutating the human STING to encode the chimpanzee residue at site 78 (78W) caused it to become resistant to cleavage by DENV NS2B3 (*Figure 2B*). Likewise, mutating the chimpanzee STING at residue 78 to the human amino acid (78R) rendered the chimpanzee STING susceptible to cleavage (*Figure 2B*). We saw no effect of mutations at a second site, 230, either alone or in combination with residue 78 (*Figure 2B*). Previously, it had been shown that STING site 78 may be important for retention in the endoplasmic reticulum (ER) (*Sun et al., 2009*). To ensure that ER retention was not disrupted by the mutations that we tested, we disrupted both copies of STING in A549 cells using CRISPR-Cas9 targeting, and then stably re-complemented them with wildtype or 78W (cleavage resistant) human STING (*Figure 2—figure supplement 1*). Both the wildtype and mutant STING similarly localized to the ER (*Figure 2—figure supplement 2*). It is logical that the 78W substitution would not affect ER-localization of STING, since 78W is naturally occurring in the chimpanzee STING protein. Therefore, we can conclude that position 78R is a critical determinant for dengue virus cleavage, either as a binding site or a cleavage site for the dengue protease. It was previously estimated that STING is cleaved in a way that divides the protein into approximately 25% and 75% of its original molecular weight, with the N-terminus of the protein representing the smaller portion (*Yu et al., 2012*). This would place the cleavage site in the vicinity of the 78th residue. In addition, the 78/79 'RG' motif is

in good agreement with what is known about the preferences of NS2B3, where glycine (G) often lies directly downstream of the peptide cleavage site, and an arginine (R) directly upstream (*Li et al., 2005a*).

Next we wished to ensure that the cleavage of STING alters its ability to signal in the interferon induction pathway. Transfection of plasmids encoding STING into cells is sufficient to activate the interferon induction pathway (*Ishikawa and Barber, 2008*). We again performed cotransfection of plasmids encoding STING and the dengue virus protease. 48 hr after transfection, cell lysates were probed in western blots for phosphorylated IRF3 (pIRF3) and for total IRF3. We found that pIRF3 was reduced when human or chimpanzee STING was susceptible to NS2B3 cleavage, and not reduced when STING was resistant to cleavage (*Figure 2C*, bottom). We also monitored the activation of the interferon-beta (IFNb) promoter. We performed an identical cotransfection assay with plasmids encoding STING and NS2B3, only in triplicate, and with two additional plasmids: one encoding a firefly luciferase reporter gene downstream of the IFNb promoter, and another encoding a renilla luciferase reporter gene downstream of a CMV promoter (used to normalize transfection efficiencies between samples, by taking the ratio of firefly:renilla luciferase). With human STING and the version of chimpanzee STING rendered sensitive to cleavage (78R), there was a significant reduction in firefly luciferase production in the presence of active NS2B3, in comparison to the catalytically dead version of the protease (*Figure 2C*, top). This reduction is not observed with chimpanzee STING, or with human STING rendered resistant to cleavage by the 78W mutation.

We then verified these results with infection experiments. We stably re-complemented our A549 STING knockout cells, using retroviral transduction, to express various forms of STING: chimpanzee or human 78W (both cleavage resistant), human or chimpanzee 78R (both cleavage susceptible), or cells were complemented with an empty vector (*Figure 2—figure supplement 1*). These cells were infected with dengue virus 2 (strain 16681) at MOI 0.3. At 24 and 48 hr post infection, supernatant was harvested and viral content was quantified by plaque assay on BHK21 cells, and at the same time cells were harvested and lysed for western blot. We found that A549 cells re-complemented with STING, regardless of the version, produced less dengue virus than the STING knockout cell line that was not re-complemented (*Figure 3*). However, cells re-complemented with a cleavage-resistant STING produced less virus than those re-complemented with a cleavage-susceptible STING (*Figure 3*). In fact, cell lines in this experiment that differ by only a single amino acid in STING demonstrate as much as a 176-fold change in infectious virus produced at 24 hr post-infection, according to the titration experiments (human versus human 78W STING). The difference remains significant at 48 hr post-infection. This suggests that cleavage of STING is critically important for dengue virus replication, and has a large impact on viral titers. The STING cleavage product was not visible in the western blots performed during these experiments. This cleavage product is typically only detectable when cells are treated with MG132 proteasome inhibitor for several hours before cell lysis. While our transfection-based cleavage assays typically incorporate MG132 treatment (see Materials and methods), it was not used in the infection experiments shown here in order to not perturb infectious virus produced. In a separate experiment performed in the presence of MG132, we do see the cleavage of STING during infections (*Figure 3—figure supplement 1*). Further, the cleavage of endogenous STING during dengue infection was previously demonstrated under other conditions (*Aguirre et al., 2012*; *Yu et al., 2012*).

We next wanted to determine if our newly identified cleavage determinant could explain the resistance to STING cleavage seen in other species. The dengue protease also cannot cleave rhesus macaque, marmoset, or mouse STING (*Figure 1B*), all of which deviate from the 'RG' motif found in human STING (highlighted green in *Figure 4A*). We next performed site-directed mutagenesis to alter either the 78th or 79th residue in STING of these species. We found that, in all cases, mutations that restored this motif to the human 'RG' restored susceptibility to cleavage (*Figure 4B*). Consistent with previous studies (*Aguirre et al., 2012*; *Yu et al., 2012*), mutation of residues 93–96 in mouse STING to match the human 'LRRG' did not confer susceptibility to cleavage by NS2B3 (*Figure 4B*). Overall, these results further support the conclusion that sites 78 and 79 are critical determinants for cleavage by the DENV NS2B3 protease. An 'RG' motif at these two positions is both necessary and sufficient to make primate and rodent STING susceptible to cleavage by the DENV2 protease.

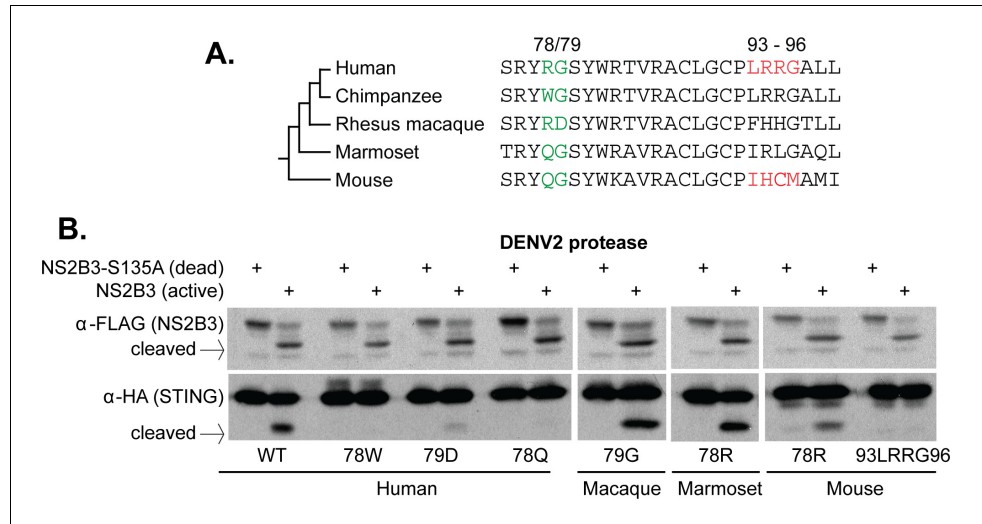

**Figure 4.** Residues 78 and 79 of STING define a dengue virus cleavage determinant in both primate and mouse STING. (A) A phylogeny and multiple sequence alignment of STING from various primate species and mouse. Shown in green is the 78/79 motif in STING that is mutated in panel B. Shown in red is the motif changed in mouse STING, only, in panel B. (B) Site directed mutagenesis was performed on human, rhesus macaque, marmoset, or mouse STING at sites 78/79 or 93–96 (mouse only). 293T cells were cotransfected with mammalian expression plasmids encoding STING along with wildtype or mutant NS2B3. 24 hr after transfection, whole-cell lysate was harvested and probed for FLAG or HA by western blot. Data presented are representative of at least two experiments.

DOI: https://doi.org/10.7554/eLife.31919.010

## The 78–79 RG motif in STING is a universal cleavage determinant for the proteases of human and sylvatic dengue viruses

To test whether these results are generalizable to other dengue viruses endemic in humans, we cloned the region encoding the NS2B3 protease complex from three additional DENV isolates (one from each endemic human virus): DENV1 (Hawaii), DENV3 (Philippines/H887/1956), and DENV4 (H241). While some of these proteases expressed better than others, all were able to cleave wildtype human STING far more efficiently than human STING bearing the 78W mutation (*Figure 5A*). This data indicates that the 78/79 RG motif of STING is recognized (i.e. bound or cleaved) by the NS2B3 proteases of all endemic human dengue viruses. If residues 78/79 in fact constitute the actual cleavage site for the protease, this would be in line with biochemical studies showing that the proteases of all four endemic human dengue viruses have similar cleavage motif preferences (*Li et al., 2005a*).

We next cloned the NS2B3 protease from a sylvatic dengue strain (DakAr-141069). This virus was first isolated from an *Ae. luteocephalus* mosquito in Senegal in 1999 (*Vasilakis et al., 2008*). We find that this viral protease also cleaves human STING, but not the STING of chimpanzee, rhesus macaque, or marmoset (*Figure 5B*). Further, the restoration of the 'RG' motif at positions 78/79 again renders all of these STING proteins susceptible to cleavage (*Figure 5B*), indicating that the sylvatic protease is targeting (i.e. binding or cleaving) the same cleavage determinant as the proteases from human dengue viruses. This is consistent with the high degree of similarity between human and sylvatic proteases, as can be seen in alignment of the two (*Figure 5—figure supplement 1*).

It is curious to find that a sylvatic dengue virus does not cleave nonhuman primate STING. Since we don't know the exact species that constitute the viral reservoir, we next considered the question of whether any nonhuman primates encode the correct cleavage determinant at positions 78/79 in STING. To address this, we harvested mRNA from cell lines derived from 16 different nonhuman primate species (see Materials and methods). From these mRNA pools, we made cDNA libraries and sequenced the STING cDNA using Sanger sequencing. We also gathered STING sequence for 14 additional primate species from Genbank. An alignment of the eight amino acid region in STING surrounding the 78/79 cleavage determinant (downward arrow in *Figure 6A*) is shown for all 30 of these primate species (a full alignment of primate STING sequences is provided in *Supplementary file 1*).

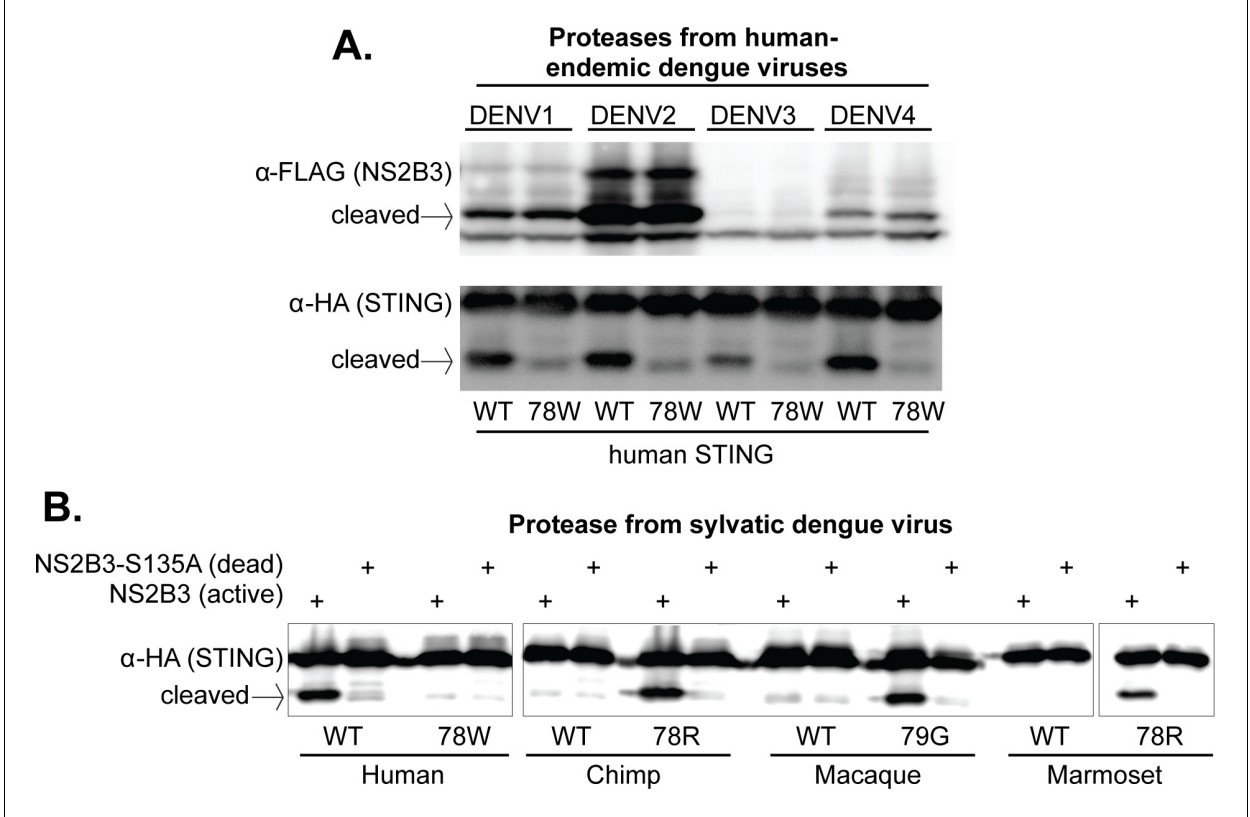

**Figure 5.** The 78/79 cleavage determinant in STING is targeted by proteases encoded by all endemic human dengue viruses, and by at least one sylvatic dengue virus. (**A**) 293T cells were cotransfected with plasmids encoding NS2B3 from DENV1-4 along with human STING with or without a mutation at site 78. Western blotting was performed on lysate harvested 24 hr post transfection to detect NS2B3 (anti-FLAG) or STING (anti-HA). Data presented are representative of at least two experiments. (**B**) 293T cells were cotransfected with plasmids encoding the indicated STING and the NS2B3 from a sylvatic isolate of dengue virus (DakAr-141069). 24 hr post transfection, lysates where harvested, run on a gel, and western blotting was performed with an anti-HA antibody to detect STING. All data presented are representative of at least two experiments.
DOI: https://doi.org/10.7554/eLife.31919.011

The following figure supplement is available for figure 5:

**Figure supplement 1.** Alignment of the NS2B3 protease from sylvatic (top) versus human (bottom) dengue viruses.
DOI: https://doi.org/10.7554/eLife.31919.012

With the exception of chimpanzees and bonobos, all apes encode the same amino acids as human in this motif, constituting the correct cleavage determinant for dengue virus. In contrast, no monkey species encodes an 'RG' at positions 78/79. Instead, Old World monkeys all encode 'RD,' which is the same motif found in the macaque clone that we have tested. Also, no monkeys from the Americas encode an 'RG' at these residues, and instead these species encode a 'QG' at positions 78/79, just like the marmoset clone tested herein.

Finally, we queried the entire Genbank database for STING sequences available from placental mammals. Mice and pigs, two current models for dengue virus infection (*Cassetti et al., 2010*), also do not have the correct RG residues at STING 78/79 (*Figure 6A*). Out of the entire database, only two other mammals were identified that share the exact same sequence as humans in the eight amino acid region surrounding the newly identified dengue virus cleavage determinant in STING: chinchilla and naked mole rat, both of which are rodents (*Figure 6A*). A third rodent species, the desert woodrat, has the RG at positions 78/79, but encodes two amino acid substitutions just downstream of these residues, compared to human STING (*Figure 6A*). The fact that only a small handful of mammals encode an RG at position 78/79, out of the entire database, may in part explain why modeling dengue viruses in animals has been so difficult. We next wished to determine if STING is in fact cleaved by dengue in these rodent species, since all three already serve as animal models for

biological research (*Keane et al., 2014*; *Nathaniel et al., 2013*; *Shimoyama et al., 2016*; *Campbell et al., 2016*; *Skopec et al., 2013*). We synthesized HA-tagged STING genes for the rodent species discussed in *Figure 6A*, as well as an additional rodent (13-lined ground squirrel) which does not have the 'RG' motif at STING 78/79, as a negative control. Cleavage assays were performed using the co-transfection assay described previously. We see that the STING of naked mole rat and desert woodrat is clearly cleaved by the DENV2 protease, in that the cleavage product is evident (*Figure 6B*). We do not see a cleavage product for chinchilla STING, even under long exposure, but we do see the STING band disappear. It's possible that, in this case, the cleavage product is too unstable to be detected. The identification of animal models encoding STING proteins that can be cleaved by dengue might be important; the advantage to using such species as models is that, unlike in STING knockout mice, the STING pathway would be intact in these animals.

## Discussion

In humans cells, sylvatic and human dengue viruses replicate similarly (*Vasilakis et al., 2007*, *2008*). These results have been interpreted to mean that there is little or no adaptive barrier for the emergence of sylvatic dengue viruses into human populations. Our data agree with, but add a new element to, this model. Rather than there being critical differences between human and sylvatic viruses, our data suggest that there are critical differences between human and monkey hosts. This difference tracks, at least in part, to STING, revealing one way in which dengue viruses are reaching higher titers in humans than in monkey models. Collectively our data suggest that all dengue viruses cleave human STING, but not the versions of STING found in most other mammals. We have used the STING proteins of closely related primate species to map the determinant of cleavage in STING. We find that the viral protease is recognizing (i.e. cutting or binding) residues around positions 78/79 of STING. We show that an 'RG' motif at these two residues is necessary and sufficient for cleavage by the proteases of all four human epidemic dengue viruses, and one sylvatic dengue virus. Yet, only some apes and three rodent species, out of all of the mammalian STING sequences in Genbank, encode an RG at positions 78/79.

Why do dengue viruses universally cleave human but not monkey STING? It's possible that what we have uncovered is a brilliant method for balancing alternate host species, one of which is dense and abundant (humans), versus others that are spare and exist in smaller populations (primates in nature). In this scenario, dengue viruses have evolved to suppress innate immunity in humans in order to increase viral titers and spread, even though this trait comes at the cost of increased pathogenicity in some individuals. This might be a good strategy in our abundant and dense host population, where the fitness cost of severe disease in a fraction of individuals would be outweighed by excellent spread. Remarkably, though, dengue viruses have achieved this by evolving to recognize a portion of human STING that is not conserved in the STING of the wild and more rare animals that serve as their sustaining reservoir in nature, allowing the viruses to maintain decreased pathogenicity in these species. The evolution of the viral proteases to cleave human STING and simultaneously to avoid cleavage of monkey STING would be expected to reduce virus titers in monkeys, as the interferon pathway would be at least partially enabled. This may be beneficial for many reasons, one of which is that the production of a low-level innate immune response may allow the virus to replicate in reservoir host species without inducing high titers and strong adaptive immune responses. Alternately, a second possibility is that sylvatic dengue viruses do cleave the STING of monkeys, but that the sylvatic virus (DakAr-141069) that we tested is not representative. However, we find this unlikely. Because DENV1-4 also cannot cleave monkey STING, and all derive from the sylvatic reservoir, this supports the finding that sylvatic viruses do not cleave monkey STING. Third, a final possibility is that apes are critical reservoirs for dengue viruses in nature. Gorillas encode 'RG' at 78/79 and are found in Africa, while orangutans and gibbons are found in Asia and also encode the correct cleavage motif for the dengue virus protease. In fact, wild orangutans have previously been found to have neutralizing antibodies against dengue virus (*Wolfe et al., 2001*). While apes could be playing a role as sylvatic hosts, the highly endangered and rare status of most apes makes it hard to believe that they are playing a major role in sustaining sylvatic dengue virus currently (*Geissmann, 2007*; *Walsh et al., 2003*).

The specific primate species that serve as the sustaining reservoir for sylvatic dengue viruses in nature are unknown (*Vasilakis et al., 2011*). Various Old World monkey species in both Asia and

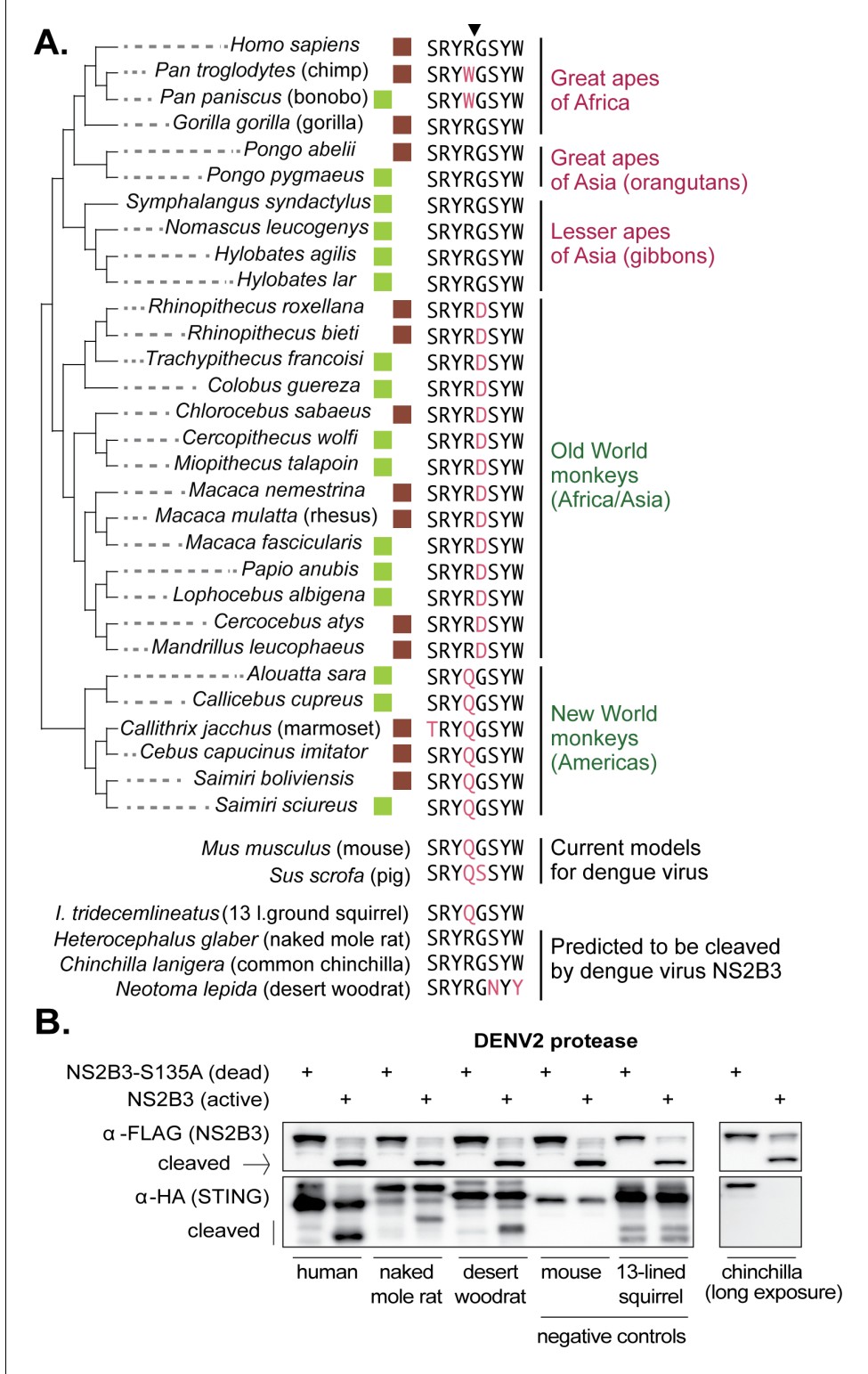

**Figure 6.** The dengue virus cleavage determinant in STING of various species. (A) An alignment of the eight amino acid region in STING surrounding residues 78R/79G, the newly identified dengue virus cleavage determinant (downward arrow at top). Deviations from the human motif are highlighted. The green boxes indicate STING orthologs sequenced as part of this study. The brown boxes indicate STING sequences obtained from Genbank. Apes are shown at the top of the tree (pink type), monkeys at the bottom (green type). Depicted below are sequences from the same region of STING from two animal models for dengue virus (mouse, pig

*Figure 6 continued on next page*

*Figure 6 continued*

[*Cassetti et al., 2010*]), several small rodent species which encode the correct cleavage motif at 78/79 (naked mole rat, common chinchilla, desert woodrat), and one that does not (13 lined ground squirrel). Genbank accession numbers of sequences shown: mouse (XP_017173483), pig (XP_005661761), 13-lined ground squirrel (XM_005327275), naked mole rat (JAO02071), chinchilla (XP_005382124), and desert woodrat (OBS58238). (**B**) STING-HA genes were synthesized for the rodent species discussed in panel A. Cleavage assays were performed by co-transfecting plasmids encoding the dengue protease (dead or active) as well as each STING, and then performing immunoblotting as described in the methods. The data presented are representative of at least two independent experiments.

DOI: https://doi.org/10.7554/eLife.31919.013

Africa are suspected hosts (*Vasilakis et al., 2011*; *Rodhain, 1991*; *Diallo et al., 2003*). For instance, sylvatic dengue viruses have been isolated directly from macaques (*Macaca fascicularis)* and leaf monkeys (*Presbytis obscura*) that were placed as sentinels in forest canopies (*Rudnick, 1986*). Other primates have been shown to have antibodies to dengue virus, including macaques (*Macaca fascicularis* and *Macaca nemestrina*), leaf monkeys (*Presbytis cristata*) (*Rudnick, 1965*), African green monkeys (*Chlorocebus sabaeus*) (*Diallo et al., 2003*), and one ape species, the Bornean orangutan (*Pongo pygmaeus*) (*Wolfe et al., 2001*). But these results are not definitive due to the cross-reactivity of antibodies directed against various flaviviruses (*Calisher et al., 1989*; *Tesh et al., 2002*; *Mansfield et al., 2011*), and the possibility that some primates might be accidental, rather than sustaining reservoir hosts (*Vasilakis et al., 2011*). Instead of being directly isolated from primates, most sylvatic dengue viruses have been obtained from forest mosquitoes (*Diallo et al., 2003*; *Rudnick, 1986*), or from humans that contracted the virus in the forest (*Pyke et al., 2016*; *Cardosa et al., 2009*; *Franco et al., 2011*). Therefore, there are many deficiencies in our understanding of the natural reservoir for dengue viruses. Interestingly, though, we have not identified any monkey species with an 'RG' at positions 78/79 in STING. Our results would indicate that dengue viruses, in general, cannot cleave the STING of monkey hosts. Our data suggests that even a sylvatic dengue virus, which we find targets the same residues in STING, would not be able to cleave STING of monkeys.

There are also implications of these findings to our understanding of dengue virus model organisms. If dengue proteases do not cleave most nonhuman forms of STING, this may at least partially explain why it has been so difficult to model dengue infection in immune-competent animals. Nonhuman primates infected with dengue virus generally don't develop clinical signs of disease, consistent with enhanced control of the virus compared to humans (*Cassetti et al., 2010*). In fact, when human dengue viruses have been observed to replicate robustly in primate cell lines, these experiments have typically been done in cells such as Vero (*Vasilakis et al., 2008*; *Rossi et al., 2012*; *Vasilakis et al., 2009*) which are deficient in the type I interferon response (*Osada et al., 2014*; *Desmyter et al., 1968*). Human dengue viruses also cannot cleave mouse STING ((*Aguirre et al., 2012*; *Yu et al., 2012*) and herein), consistent with the heightened control of this virus in mice as well (*Cassetti et al., 2010*). Dengue virus will replicate to high titers in mice lacking key genes important for the interferon response, but for many reasons it is desirable to develop animal models in immune competent hosts (*Cassetti et al., 2010*). STING now adds to a growing list of host proteins that regulate viral infection differently even in closely related host species (for example, [*Stabell et al., 2016*; *Lou et al., 2016*; *Rowley et al., 2016*; *Kerr et al., 2015*; *Meyerson et al., 2015*; *Demogines et al., 2012*; *Stremlau et al., 2004*; *Demogines et al., 2013*; *Ng et al., 2015*; *Hueffer et al., 2003*; *Radoshitzky et al., 2008*; *Patel et al., 2012*; *Elde et al., 2009*; *Martin et al., 2013*; *Miller et al., 2012*; *Sawyer and Elde, 2012*; *Meyerson et al., 2017*]). The identification of such genes is critical to our understanding of viral adaptation during host switching, and to the development of animal models in which to study human viruses.

It is possible that the identification of small mammals that have a cleavage-susceptible STING would facilitate the development of better animal models for studying dengue virus. Our work suggests the identity of three such species: the naked mole rat, the common chinchilla, and the desert woodrat. All three of these small rodents are already used as animal models in biomedical research, and the genomes of all three have been sequenced (*Keane et al., 2014*; *Nathaniel et al., 2013*; *Shimoyama et al., 2016*; *Campbell et al., 2016*; *Skopec et al., 2013*). These species could be

superior to STING knockout mice, in that the STING pathway would be intact and the cleavage of STING by the virus would be naturally modeled rather than just bypassed. These species may also be superior to future models where mouse (*Mus musculus*) STING would be replaced with human STING in transgenic animals. In this case, it is unknown if human STING would perform all of its functions the same in mouse as it does in humans. The advantage of using a rodent model with a STING that is naturally susceptible to dengue virus cleavage would be that the STING pathways would all be fully functional and intact. It is important to point out that, in addition to cleaving STING, dengue viruses modulate the interferon response in other ways as well. For instance, dengue viruses also bypass the type I interferon response by binding and degrading host STAT2 via the viral NS5 protein (*Ashour et al., 2009*; *Jones et al., 2005*; *Mazzon et al., 2009*; *Best, 2017*). Ideally, human dengue viruses would also be able to bind and degrade STAT2 in newly developed models, as they do in humans. Further, dengue viruses neutralize both the type I and type II interferon responses in other ways as well (*Perry et al., 2011*; *Shresta et al., 2004*; *Aguirre et al., 2017*; *Aguirre and Fernandez-Sesma, 2017*). Other known host-virus interactions would also need to be characterized in any potential new model organism.

It is notable that chimpanzees and bonobos encode STINGs that are resistant to cleavage, while STINGs of all other apes are susceptible. These two species differ from other apes in encoding a 'WG' at 78/79 of STING rather than the 'RG' encoded by all other apes (*Figure 6*). Remarkably, it was previously found that the 'W' at position 78, destroying the dengue cleavage determinant, was fixed by positive natural selection in wild chimpanzee populations (*Mozzi et al., 2015*). Chimpanzees are not one of the suspected natural reservoirs of dengue virus, but chimpanzee ranges do co-occur with known human outbreaks and with sylvatic cycles (*Figure 1—figure supplement 1*). One model is that, as dengue virus spread through Africa, a SNP in chimpanzee STING (or the STING of the chimpanzee/bonobo ancestor) at position 78 started to experience strong selection because it provided protection against cleavage by dengue viruses. This would have driven a selective sweep in chimpanzee populations, causing this species to become less susceptible to viral infection. It has previously been proposed that evolutionary pressure imposed by flavivirus proteases can drive selection at cleavage sites. For examples, the hepatitis C protease cleaves MAVS, another host signaling protein in the interferon induction cascade (*Li et al., 2005b*; *Meylan et al., 2005*). MAVS has experienced positive selection at a residue in the cleavage site for the hepatitis C virus protease (*Patel et al., 2012*). The authors of this study speculated that ancient viruses may have exerted selective pressure on primate genomes to acquire mutations in the cleavage site.

Like other viruses, dengue viruses remodel their host cellular environment in numerous ways, including the cleavage of STING and degradation of STAT2. Using the rich information that exists on how dengue viruses accomplish this, the genetic susceptibility of both suspected reservoir hosts, and potential new animal models, can be systematically assessed. Characterizing how host-virus interactions play out uniquely in different host species will help us to understand dengue virus in critical ways. For instance, it will reveal how dengue viruses do (or do not) need to evolve their genomes as they transmit to humans from nonhuman primates in nature. Also, understanding the genetics of host tropism will help identify better laboratory animals that can be used to study dengue virus pathogenesis and to develop drugs and vaccines.

# Materials and methods

**Key resources table**

| Reagent type (species) or resource | Designation | Source or reference | Identifiers | Additional information |
|---|---|---|---|---|
| gene (Homo sapiens) | STING; TMEM173 | NA | GENBANK:NM_198282 | GENBANK:MF622062 |
| gene (Homo sapiens) | STING; TMEM173 | this study | | GENBANK:MF622062 |
| gene (Pan troglodytes) | STING; TMEM173 | NA | GENBANK:XM_016953921 | |
| gene (Pan paniscus) | STING; TMEM173 | this study | | GENBANK:MF616339 |
| gene (Gorilla gorilla) | STING; TMEM173 | NA | GENBANK:XM_0040426 | |
| gene (Pongo abelii) | STING; TMEM173 | NA | GENBANK:XM_002815952 | |

*Continued on next page*

*Continued*

| Reagent type (species) or resource | Designation | Source or reference | Identifiers | Additional information |
|---|---|---|---|---|
| gene (Hylobates agilis) | STING; TMEM173 | this study | | GENBANK:MF616342 |
| gene (Symphalangus syndactylus) | STING; TMEM173 | this study | | GENBANK:MF616343 |
| gene (Nomascus leucogenys) | STING; TMEM173 | this study | | GENBANK:MF616344 |
| gene (Hylobates lar) | STING; TMEM173 | this study | | GENBANK:MF616341 |
| gene (Rhinopithecus roxellana) | STING; TMEM173 | NA | GENBANK:XM_010388119 | |
| gene (Rhinopithecus bieti) | STING; TMEM173 | NA | GENBANK:XM_017895026 | |
| gene (Trachypithecus francoisi) | STING; TMEM173 | this study | | GENBANK:MF616352 |
| gene (Colobus guereza) | STING; TMEM173 | this study | | GENBANK:MF616351 |
| gene (Chlorocebus sabaeus) | STING; TMEM173 | NA | GENBANK:XM_008014636 | |
| gene (Cercopithecus wolfi) | STING; TMEM173 | this study | | GENBANK:MF616350 |
| gene (Miopithecus talapoin) | STING; TMEM173 | this study | | GENBANK:MF616349 |
| gene (Macaca nemestrina) | STING; TMEM173 | NA | GENBANK:XM_011716377 | |
| gene (Macaca mulatta) | STING; TMEM173 | NA | GENBANK:XM_015141010 | |
| gene (Macaca mulatta) | STING; TMEM173 | this study | | GENBANK:MF622060 |
| gene (Macaca fascicularis) | STING; TMEM173 | this study | | GENBANK:MF616346 |
| gene (Papio papio) | STING; TMEM173 | this study | | GENBANK:MF616348 |
| gene (Lophocebus albigena) | STING; TMEM173 | this study | | GENBANK:MF616347 |
| gene (Cercocebus atys) | STING; TMEM173 | NA | GENBANK:XM_012090448 | |
| gene (Mandrillus leucophaeus) | STING; TMEM173 | NA | GENBANK:XM_011997224 | |
| gene (Aloutta sara) | STING; TMEM173 | this study | | GENBANK:MF616355 |
| gene (Callicebus cupreus) | STING; TMEM173 | this study | | GENBANK:MF616354 |
| gene (Callithrix jacchus) | STING; TMEM173 | NA | GENBANK:XM_00898588 | |
| gene (Callithrix jacchus) | STING; TMEM173 | this study | | GENBANK:MF622061 |
| gene (Cebus capucinus imitator) | STING; TMEM173 | NA | GENBANK:XM_017536735 | |
| gene (Samiri boliviensis) | STING; TMEM173 | NA | GENBANK:XM_003933913 | |
| gene (Saimiri sciureus) | STING; TMEM173 | this study | | GENBANK:MF616353 |
| gene (Mus musculus) | STING; TMEM173 | NA | GENBANK:NM_001289591 | |
| gene (Sus scrofa) | STING; TMEM173 | NA | GENBANK:XP_005661761 | |
| gene (Heterocephalus glaber) | | NA | GENBANK:JAO02071 | |
| gene (Chinchilla lanigera) | | NA | GENBANK:XP_005382124 | |
| gene (Neotoma lepida) | | NA | GENBANK:OBS58238 | |
| gene (Dengue viurs 2) | NS2B3 | NA | GENBANK:M29095 | |
| cell line (Homo sapiens) | 293T cells | ATCC | CRL-3216 | |

*Continued on next page*

*Continued*

| Reagent type (species) or resource | Designation | Source or reference | Identifiers | Additional information |
|---|---|---|---|---|
| cell line (Homo sapiens) | A549 cells | ATCC | CCL-185 | |
| antibody | Rat anti-HA-HRP (3F10) | Sigma | 11867423001 | |
| antibody | Mouse anti-Flag (M2) | Sigma | F3165 | |
| antibody | Rabbit anti-pIRF3 | abcam | ab76493 | |
| antibody | Rabbit anti-IRF3 | Santa Cruz Biotech | sc-9082 | |
| antibody | Rabbit anti-GAPDH | Cell Signaling | 14C10 | |
| antibody | Rabbit anti-STING | abcam | ab92605 | |
| antibody | Mouse anti-Actin (C4) | Santa Cruz Biotech | Sc47778 | |
| recombinant DNA reagent | DENV2 NS2B3 WT (plasmid) | PMID: 1642612 | | Progenitors: DENV2 NGC (GENBANK:M29095), pCR3.1 |
| recombinant DNA reagent | DENV2 NS2B3 S135A (plasmid) | PMID: 1642612 | | Progenitors: DENV2 NS2B3 WT pCR3.1 plasmid, SDM |
| recombinant DNA reagent | DENV1 (Hawaii) cDNA | this paper | | Progenitors: World Reference Center for Emerging Viruses and Arboviruses (WRCEVA) Catalog number NR-4287 |
| recombinant DNA reagent | DENV2 (New Guinea C) cDNA | this paper | | Progenitors: World Reference Center for Emerging Viruses and Arboviruses (WRCEVA) Catalog number NR-4288 |
| recombinant DNA reagent | DENV3 (Philippines/H87/1956) cDNA | this paper | | Progenitors: World Reference Center for Emerging Viruses and Arboviruses (WRCEVA) Catalog number NR-2771 |
| recombinant DNA reagent | DENV4 (H241) cDNA | this paper | | Progenitors: World Reference Center for Emerging Viruses and Arboviruses (WRCEVA) Catalog number NR-4289 |
| recombinant DNA reagent | DENV1 (Hawaii) NS2B3 WT (plasmid) | this paper | | Progenitors: DENV1 (Hawaii) cDNA, pCR3.1 |
| recombinant DNA reagent | DENV2 (New Guinea C) NS2B3 WT (plasmid) | this paper | | Progenitors: DENV2 (New Guinea C) cDNA, pCR3.1 |
| recombinant DNA reagent | DENV3 (Philippines/H87/1956) NS2B3 WT (plasmid) | this paper | | Progenitors: DENV3 (Philippines/H87/1956) cDNA, pCR3.1 |
| recombinant DNA reagent | DENV4 (H241) NS2B3 WT (plasmid) | this paper | | Progenitors: DENV4 (H241) cDNA, pCR3.1 |
| recombinant DNA reagent | Sylvatic (DakAr-141069) Dengue NS2B3 Protease (WT) | this paper | | Progenitors: DakAr-141069 NS2B3 sequence (GenBank EF105389) |
| recombinant DNA reagent | Sylvatic (DakAr-141069) Dengue NS2B3 Protease (S135A) | this paper | | Progenitors: Sylvatic (DakAr-141069) Dengue NS2B3 Protease (WT) SDM product |
| recombinant DNA reagent | human cDNA | this paper | | Progenitors: A549 cell line (ATCC CCL-185) |
| recombinant DNA reagent | chimpanzee cDNA | this paper | | Progenitors: Coriell PR00748 |
| recombinant DNA reagent | rhesus macaque cDNA | this paper | | Progenitors: Mm265-95 |
| recombinant DNA reagent | marmoset cDNA | this paper | | Progenitors: Coriell PR07404 |
| recombinant DNA reagent | mouse cDNA | this paper | | Progenitors: RNA extracted from mouse liver |

*Continued on next page*

*Continued*

| Reagent type (species) or resource | Designation | Source or reference | Identifiers | Additional information |
|---|---|---|---|---|
| recombinant DNA reagent | human STING-HA (plasmid) | this paper | | Progenitors: human cDNA, pcDNA3.1 plasmid |
| recombinant DNA reagent | human STING-HA (plasmid) | this paper | | Progenitors: human cDNA, pLPCX plasmid |
| recombinant DNA reagent | human STING(R78W)-HA (plasmid) | this paper | | Progenitors: human STING-HA pcDNA3.1 SDM product |
| recombinant DNA reagent | human STING(R78W)-HA (plasmid) | this paper | | Progenitors: human STING-HA pLPCX SDM product |
| recombinant DNA reagent | human STING(R79D)-HA (plasmid) | this paper | | Progenitors: human STING-HA pcDNA3.1 SDM product |
| recombinant DNA reagent | human STING(R78Q)-HA (plasmid) | this paper | | Progenitors: human STING-HA pcDNA3.1 SDM product |
| recombinant DNA reagent | chimpanzee STING-HA (plasmid) | this paper | | Progenitors: chimpanzee cDNA, pcDNA3.1 plasmid |
| recombinant DNA reagent | chimpanzee STING-HA (plasmid) | this paper | | Progenitors: chimpanzee cDNA, pLPCX plasmid |
| recombinant DNA reagent | chimpanzee STING(W78R)-HA (plasmid) | this paper | | Progenitors: chimpanzee STING-HA pcDNA3.1 SDM product |
| recombinant DNA reagent | chimpanzee STING(W78R)-HA (plasmid) | this paper | | Progenitors: chimpanzee STING-HA pLPCX SDM product |
| recombinant DNA reagent | rhesus macaque STING-HA (plasmid) | this paper | | Progenitors: rhesus macaque cDNA, pcDNA3.1 plasmid |
| recombinant DNA reagent | rhesus macaque STING(D79G)-HA (plasmid) | this paper | | Progenitors: rhesus macaque STING-HA pcDNA3.1 SDM product |
| recombinant DNA reagent | marmoset STING-HA (plasmid) | this paper | | Progenitors: marmoset cDNA, pcDNA3.1 plasmid |
| recombinant DNA reagent | marmoset STING(Q78R)-HA (plasmid) | this paper | | Progenitors: marmoset STING-HA pcDNA3.1 SDM product |
| recombinant DNA reagent | mouse STING-HA (plasmid) | this paper | | Progenitors: mouse cDNA, pcDNA3.1 plasmid |
| recombinant DNA reagent | mouse STING(Q78R)-HA (plasmid) | this paper | | Progenitors: mouse STING-HA pcDNA3.1 SDM product |
| recombinant DNA reagent | mouse STING(93LRRG96)-HA (plasmid) | this paper | | Progenitors: mouse STING-HA pcDNA3.1 SDM product |
| recombinant DNA reagent | human STING-3xFLAG (plasmid) | this paper | | Progenitors: human cDNA, pLPCX plasmid |
| recombinant DNA reagent | human STING(R78W)-3xFLAG (plasmid) | this paper | | Progenitors: human STING-3xFLAG pLPCX SDM product |
| recombinant DNA reagent | human STING(G230A)-3xFLAG (plasmid) | this paper | | Progenitors: human STING-3xFLAG pLPCX SDM product |
| recombinant DNA reagent | human STING (R78W, G230A)-3xFLAG (plasmid) | this paper | | Progenitors: human STING-3xFLAG pLPCX SDM product |

*Continued on next page*

*Continued*

| Reagent type (species) or resource | Designation | Source or reference | Identifiers | Additional information |
|---|---|---|---|---|
| recombinant DNA reagent | chimpanzee STING-3xFLAG (plasmid) | this paper | | Progenitors: chimpanzee cDNA, pLPCX plasmid |
| recombinant DNA reagent | chimpanzee STING (W78R)-3xFLAG (plasmid) | this paper | | Progenitors: chimpanzee STING-3xFLAG pLPCX SDM product |
| recombinant DNA reagent | chimpanzee STING(A230G)-3xFLAG (plasmid) | this paper | | Progenitors: chimpanzee STING-3xFLAG pLPCX SDM product |
| recombinant DNA reagent | chimpanzee STING(W78R, A230G)-3xFLAG (plasmid) | this paper | | Progenitors: chimpanzee STING-3xFLAG pLPCX SDM product |
| recombinant DNA reagent | IFN-ß1-luc (plasmid) | PMID: 21512573 | | |
| recombinant DNA reagent | pRL-CMV (plasmid) | Promega: AF025843 | | Progenitors: pRL-null |
| commercial assay or kit | Dual-Glo Luciferase Assay System | Promega | Cat#E2920 | |
| commercial assay or kit | Superscript III First-Strand Synthesis System | Thermo Scientific | Cat#18080051 | |
| software, algorithm | MEGA7 | | | http://www.megasoftware.net/ |
| software, algorithm | ImageJ version 1.43u | | | http://rsb.info.nih.gov/ij |
| software, algorithm | Python 2.7.11 | | | https://www.python.org |
| software, algorithm | Sequencher | | | https://www.genecodes.com |

## Plasmids

DENV2 NS2B3, expressed from the pCR3.1 plasmid, was a gift from Yi-Ling Lin. This plasmid, and all DENV1-4 protease-expressing plasmids described below, include a 3x FLAG tag at the C-terminus of NS3. For the experiment where proteases from human dengue viruses DENV1-4 are compared, primers were designed at the 5' end of NS2B

(DENV1: taagcaAAGCTTcaccATGAGTTGGCCCCTC,
DENV2: taagcaAAGCTTcaccATGAGCTGGCCACTAAATGA,
DENV3: taagcaAAGCTTcaccATGAGCTGGCCACTG,
DENV4: taagcaAAGCTTcaccATGTCTTGGCCCCTTAAC) and the 3' end of NS3
(DENV1: TGCTTAgtcgacaTCTTCTTCCTGCTGCAAACTCTTTAAACTC,
DENV2: TGCTTAgtcgacaCTTTCTTCCAGCTGCAAACTCCTTG,
DENV3: TGCTTAgtcgacaCTTTCTGCCAGCTGCAAAATCCTTG,
DENV4: TGCTTAgtcgacaCTTTCTTCCACTGGCAAACTCCTTG) to amplify the NS2B + NS3 genomic region in one fragment. In this experiment, the protease from DENV2 was re-cloned so that the structure of the four protease clones was identical in all four cases. The PCR templates were cDNAs created from RNA obtained through the World Reference Center for Emerging Viruses and Arboviruses (WRCEVA) (Cat# NR-32847). DENV1 (Hawaii, NR-4287), DENV2 (New Guinea C, NR-4288), DENV3 (Philippines/H87/1956, NR-2771), and DENV4 (H241, NR-4289). The PCR products, and the plasmid containing the DENV2 protease mentioned above (gift from Yi-Ling Lin), were both digested with HindIII and Sal1. The PCR products were ligated into this plasmid and transformed into DH5α chemically competent *E.coli*. The sylvatic NS2B3 (DakAr141069) was synthesized (without an epitope tag) using the sequence information deposited on NCBI (Genbank accession EF105389). STING genes used for functional analysis were amplified from cDNA libraries constructed from the following cell lines: human (A549), chimpanzee/bonobo (STING sequence identical for these two species, clone amplified from Coriell, PR00748), rhesus macaque (Mm265-95, a gift from Welkin Johnson), marmoset (Coriell, PR07404), and mouse (generated from RNA extracted from whole liver). Either

an HA or 3xFlag tag were engineered onto the 3' end of the gene sequences, separated from the coding sequence by a 3xGlycine-Alanine (GAGAGA) linker region (nucleotide sequence = GGTGC TGGTGCTGGTGCT). These sequences were cloned into the pcDNA3.1 expression vector with a 5' Kozak sequence (GCCACC). Rodent STING constructs were synthesized (Quintarabio) to include a Kozak sequence, C-terminal HA-tag, and flanking linkers that were used for Gibson cloning into the pLPCX mammalian expression plasmid.

## STING cleavage assays

293 T cells (mycoplasma negative) were grown at 37°C in DMEM supplemented with 10% FBS, Pen/Strep, and L-glutamine. 24 hr prior to transfection, cells were plated at a density of $4.5 \times 10^5$ cells per well in a 12-well dish in antibiotic free media. Wells were transfected with 800 ng plasmid encoding STING and 800 ng plasmid encoding NS2B3 using TransIT 293 reagent (Mirus MIR 2704). For most experiments (*Figures 1B*, *3B*, *5A and B*), cells were treated with 10uM MG132 for 8 hr prior to harvesting for western blot.

## Western blotting

Cells were lysed in RIPA buffer supplemented with protease inhibitor (Roche, 4693159001). Protein concentration was calculated using the Bradford method. 10% 37.5:1 Acrylamide/Bisacrylamide gels were used to run 30 ug of whole cell lysate for each sample. Protein was transferred overnight at 30 volts onto a polyvinyl membrane. Blocking was performed with a 10% milk solution in tris-buffered saline supplemented with 0.1% TWEEN20. Primary antibodies used were used against HA (3f10 clone Sigma 11867423001), Flag (M2 clone Sigma F3165), GAPDH (CellSignaling 14C10), STING (Abcam 92605), actin (Santa Cruz Biotech Sc47778), and dengue virus NS3 (mouse polyclonal antibody raised against purified full-length NS3 from dengue 2 strain 16681 [*Heaton et al., 2010*]). Secondary antibodies used were goat-anti-mouse-HRP (Thermo 62–6520) and goat-anti-rabbit (Thermo 65–6120). Blots were developed using ECL Prime (Amersham RPN2232) and imaged using ImagQuant LAS 4000 (Amersham 28-9558-10).

## CRISPR-Cas9 mediated disruption of STING, and stable re-complementation with primate orthologs

A549 cells (mycoplasma negative) were transfected with the pSPCAS9(BB)-P2A-eGFP (PX458) with the guide RNA sequence 5' AGAGCACACTCTCCGGTACC 3'. GFP-positive cells were single-cell sorted into a 96-well dish and colonies were grown up. Cloned A549 cells were screened for homozygous mutations that disrupted the coding sequence of STING as follows. 10,000 cells were used to prep whole genomic DNA. The region surrounding the guide RNA was amplified using the following primers: 5' GTCCCCAAGGGTTCTTGGTT 3' and 5' AACCAGTCCCACTCCCAGTA 3'. Amplified genomic DNA was Sanger sequenced to determine the nature of the CRISPR-CAS9-mediated genomic disruption. A cell line with confirmed homozygous disruption of STING (*Figure 2—figure supplement 1*) was then re-complemented with primate orthologs of STING. Four different C-terminally HA-tagged versions of STING were cloned into the pLPCX retroviral vector: wildtype human STING, R78W human STING, wildtype chimpanzee STING, and W78R chimpanzee STING. These were packaged into retroviral particles by cotransfecting into 293T cells (mycoplasma negative) each pLPCX-STING construct with plasmids expressing NB-tropic murine leukemia virus (MLV) Gag-Pol and VSV-G. As a control, we also made virus to complement with an empty pLPCX vector. Supernatants were collected and used to transduce 105 A549 cells in the presence of 10 ug/mL polybrene. 24 hr post transduction, cells were selected in 0.75 ug/mL puromycin.

## Immunofluorescence

24 hr after plating, cells were fixed with 4% paraformaldehyde and permeabilized with 1% TritonX100 in PBS. Blocking was performed with 3% BSA solution in PBS. Primary antibodies used were rabbit-anti-GRP78 (BiP) (Abcam ab21685) and mouse-anti-HA (clone 16B12 abcam ab130275). Secondary antibodies used were donkey-anti-rabbit conjugated to AlexaFluor594 (Invitrogen A21207) and donkey-anti-mouse conjugated to AlexaFluor488 (Invitrogen A21202). Cells were mounted using VECTASHIELD hardset mounting media (VectorLabs H-1400).

## Dengue infection assays

The indicated STING knockout and re-complemented cell lines were plated out in F-12K media with 10% FBS, after 24 hr the cells were counted. An MOI of 0.3 was calculated for each well and dengue virus 2 (16681) was allowed to attach to cells for 1 hr at room temperature. Unattached virus was then removed from cells, 2%serum in F-12K media was added to cells and they were maintained at 37°C with 5% $CO_2$. After 24 and 48 hr the virus supernatant was removed for downstream titration on BHK21 cells. At the same time, cells were removed for downstream western blotting.

## Sequencing STING from other primate species

The following STING sequences were collected from GenBank: chimpanzee (*Pan troglodytes*, XM_016953921.1), gorilla (*Gorilla gorilla gorilla*, XM_004042612.1), Sumatran orangutan (*Pongo abelii*, XM_002815952.2), golden snub-nosed monkey (*Rhinopithecus roxellana*, XM_010388119.1), black snub-nosed monkey (*Rhinopithecus bieti*, XM_017895026.1), African green monkey (*Chlorocebus sabaeus*, XM_008014636.1), pigtail macaque (*Macaca nemestrina*, XM_011716377.1), rhesus macaque (*Macaca mulatta*, XM_015141010.1), sooty mangabey (*Cercocebus atys*, XM_012090448.1), drill (*Mandrillus leucophaeus*, XM_011997224.1), marmoset (*Callithrix jacchus*, XM_00898588.2), capuchin monkey (*Cebus capucinus imitator*, XM_017536735.1), black-capped squirrel monkey (*Saimiri boliviensis*, XP_003933962.1). The remaining STING gene sequences were obtained by direct sequencing of cDNA libraries produced from the following primary or immortalized primate fibroblast cell lines: Bonobo (*Pan* paniscus, Coriell PR00748), Bornean orangutan (*Pongo pygmaeus*, Coriell PR00650), white-handed gibbon (*Hylobates lar*, Coriell PR01131), agile gibbon (*Hylobates agilis*, Coriell PR00773), siamang (*Symphalagus syndactylus*, Coriell PR00722), white-cheeked gibbon (*Nomascus leucogenys*, Coriell PR01037), leaf monkey (*Trachypithecus francoisi*, Coriell PR01099), colobus monkey (*Colobus guereza*, Coriell PR00980), Wolf's guenon (*Cercopithecus wolfi*, Coriell PR01241), talapoin (*Miopithecus talapoin*, Coriell PR00716), crab-eating macaque (*Macaca fasicularis*, 103–06, gift from Welkin Johnson), olive baboon (*Papio anubis*, Coriell PR00978), grey-cheeked mangabey (*Lophocebus albigena*, Coriell PR01215), Bolivian red howler monkey (*Alouatta sara*, Coriell PR00708), red titi monkey (*Callicebus* (or *Plecturocebus*) *cupreus*, Coriell PR00793), common squirrel monkey (*Saimiri sciureus*, Coriell PR00603). Briefly, cells were grown in DMEM (Cellgro) supplemented with 15% FBS (Gibco) at 37°C and 5% CO2. RNA was extracted using the AllPrep DNA/RNA extraction kit (QIAGEN). cDNA libraries were generated using SuperScript III first strand synthesis kit (Invitrogen). PCR was performed using PCR SuperMix High Fidelity (Invitrogen). PCR products were directly sequenced. Each primate sequence was used as a query to search the human genome, and human STING gene was returned as the top hit. STING gene sequences generated in this study have been deposited in GenBank (accession numbers MF616339-MF616355).

## Acknowledgements

We thank Kartik Chandran, Nels Elde, Emily Feldman, Maryska Kaczmarek, Michael Smallegan, Cody Warren, and Qing Yang for thoughtful comments on this study. This work was supported by National Institutes of Health grant R01-GM-093086 to SLS. ACS was supported by an M.D./Ph.D. training fellowship from the National Institutes of Health (F30-AI-112277). NRM was supported by a graduate fellowship from the National Science Foundation, and a PDEP award from the Burroughs Wellcome Fund. RCG and RP were supported by the department of Microbiology, Immunology and Pathology at CSU. SLS is a Burroughs Wellcome Investigator in the Pathogenesis of Infectious Disease.

## Additional information

### Funding

| Funder | Grant reference number | Author |
| --- | --- | --- |
| National Institutes of Health | R01-GM-093086 | Sara L Sawyer |
| Burroughs Wellcome Fund | PATH | Sara L Sawyer |
| National Institutes of Health | F30-AI-112277 | Alex C Stabell |

| | | |
|---|---|---|
| National Science Foundation | GRFP | Nicholas R Meyerson |
| Burroughs Wellcome Fund | PDEP | Nicholas R Meyerson |

The funders had no role in study design, data collection and interpretation, or the decision to submit the work for publication.

## Author contributions

Alex C Stabell, Conceptualization, Formal analysis, Investigation, Methodology, Writing—original draft; Nicholas R Meyerson, Rushika Perera, Validation, Investigation, Writing—review and editing; Rebekah C Gullberg, Validation, Investigation, Methodology, Writing—review and editing; Alison R Gilchrist, Kristofor J Webb, Formal analysis, Investigation, Methodology; William M Old, Formal analysis, Supervision; Sara L Sawyer, Conceptualization, Data curation, Supervision, Funding acquisition, Writing—review and editing

## Author ORCIDs

Sara L Sawyer http://orcid.org/0000-0002-6965-1085

## Decision letter and Author response

Decision letter https://doi.org/10.7554/eLife.31919.021
Author response https://doi.org/10.7554/eLife.31919.022

# Additional files

## Supplementary files

• Supplementary file 1. Alignment of primate STING proteins.
DOI: https://doi.org/10.7554/eLife.31919.014

• Transparent reporting form
DOI: https://doi.org/10.7554/eLife.31919.015

## Major datasets

The following dataset was generated:

| Author(s) | Year | Dataset title | Dataset URL | Database, license, and accessibility information |
|---|---|---|---|---|
| Stabell A | 2018 | outbreak_mapping | http://dx.doi.org/10.5061/dryad.pm7ch | Available at Dryad Digital Repository under a CC0 Public Domain Dedication |

The following previously published dataset was used:

| Author(s) | Year | Dataset title | Dataset URL | Database, license, and accessibility information |
|---|---|---|---|---|
| Messina JP, Brady OJ, Pigott DM, Brownstein JS, Hoen AG, Hay SI | 2014 | Dengue_Occurrence_12122013.xlsx | https://figshare.com/articles/Dengue_Occurrence_12122013.xlsx/1035043 | Available at figshare under a CC0 Public Domain licence |

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
