## [Decision Letter]

Thank you for submitting your article "Dengue viruses cleave STING in humans but not in their nonhuman primate reservoirs" for consideration by *eLife*. Your article has been reviewed by three peer reviewers, one of whom is a member of our Board of Reviewing Editors and the evaluation has been overseen by Wenhui Li as the Senior Editor. The reviewers have opted to remain anonymous.

The reviewers have discussed the reviews with one another and the Reviewing Editor has drafted this decision to help you prepare a revised submission.

Summary:

The reviewers all feel that this is an interesting manuscript examining the ability of the dengue virus protease to cleave STING in a species-specific manner to evade induction of IFN responses. Most of the primates, including those that are a reservoir for DENV, have a STING unable to be cleaved by DENV. You have identified that it is due to a single amino acid change, which you interpret to be a new cleavage site in STING of the dengue protease. Before the manuscript is considered further for *eLife*, it is essential to provide biochemical evidence that this is truly the authentic cleavage site. Furthermore, your speculation about the evolution of the ability of the dengue protease to cleave STING is interesting, but the reviewers feel that alternative models need to be considered in the manuscript.

Essential revisions:

Reviewer #1:

This manuscript provides interesting results showing that the dengue virus infection can promote the cleavage of human STING and certain ape STING molecules but not many of the other primate STINGs. The research identified the human sequence needed for cleavage as 78RG79, which was also found in the ape proteins. Alteration of primate molecules to RG also allowed cleavage, and the authors conclude that this is the cleavage site. The results are very important in understanding the ability of dengue virus to infect humans and different non-human primate species.

The conclusion that they have identified a new cleavage site seems to be premature in the absence of any biochemical evidence. The authors need to provide biochemical evidence for cleavage at this site or alter their conclusions to state that they have identified sequences that are required.

Second, they speculate in the abstract and elsewhere that "dengue viruses have evolved to fine-tune their pathogenicity in primates, but not yet in humans." I understand the evolution of a parasite to cause limited pathogenicity in its host to allow host survival, but where did the ability to cleave human STING come from? Was this from infection of an as yet unidentified species or did it evolve with infection of humans? This seems to be a very incomplete model.

Reviewer #2:

This is an interesting manuscript that investigates in more detail the ability of the protease of DENV to cleave STING in a species-specific manner to evade induction of IFN responses. The authors find that most of the primates, including those that are a reservoir for DENV, have a STING unable to be cleaved by DENV. They found that this is due to a single amino acid change, most likely representing the cleavage site in STING of the dengue protease. In general, the data are consistent with this conclusion. However, some of the conclusions of the authors are too farfetched and they should be tempered.

1) The authors are likely right in the cleavage sequence of STING, as one single amino acid change make human STING not cleaved and primate or mouse STING cleaved. However, there is always the possibility that this sequence participates in recognition by the protease and that cleavage occurs somewhere else. In the absence of direct evidence of the cleavage site, the authors should refer to this sequence as required for cleavage, but they cannot exclude that cleavage occurs at a different site.

2) The experiments in Figure 3 clearly show that a single amino acid at STING determines the efficiency of inhibition of DENV by STING. Not shown in this figure is the number of DENV infected cells. This is important, as the authors did not see differences in levels of STING, which most likely indicate that the majority of the cells are not infected and therefore, that STING is not cleaved in most of the cells due to the lack of infection. In order to proof that STING is cleaved in DENV infected cells, the authors should sort DENV infected versus non-infected cells and determine the amount of STING in both populations as compared with a control host protein.

3) The speculation of the authors: "We propose that dengue viruses have evolved to fine-tune their pathogenicity in their monkey reservoir hosts, but not yet humans. Reduced viral pathogenicity within reservoir hosts is predicted to be beneficial to both host and virus" does not make sense. This will mean that the protease of DENV just by chance happened to cleave human STING, which also just by chance happened to be a major viral sensor. An alternative most likely explanation is that humans are since many thousands of years the main host for propagation of dengue virus, and therefore, the protease of DENV has evolved to cleave STING of the main host, humans, sacrificing the ability to cleave STING in primates, which are minor hosts, and therefore contribute less to the overall burden of dengue virus in nature. In other words, between two DENV, one cleaving human STING and the other cleaving primate STING, the one cleaving human STING has a competitive advantage over the one cleaving primate STING, as the virus has been mainly adapted to human and human mosquitoes, and primates might now be spillover reservoirs, but not main drivers of DENV evolution.

Reviewer #3:

It has been demonstrated that the DENV protease can cleave Human STING. Using an evolution-guided approach, the authors determine the ability of the DENV protease to cleave STING varies across species including primates due to sequence differences at amino acid positions 78 and 79. This finding is significant because it answers the long-standing question of why dengue does not replicate to high titers and display pathology similar to what is observed during human infection in primate models. I feel this is a thorough study, in particular, due to the use of genetic complementation, and the various STING orthologs and DENV (1-4) proteases assayed.

Some modifications should be made to the text including the Title.

---

## [Author Response]

Summary:The reviewers all feel that this is an interesting manuscript examining the ability of the dengue virus protease to cleave STING in a species-specific manner to evade induction of IFN responses. Most of the primates, including those that are a reservoir for DENV, have a STING unable to be cleaved by DENV. You have identified that it is due to a single amino acid change, which you interpret to be a new cleavage site in STING of the dengue protease. Before the manuscript is considered further for eLife, it is essential to provide biochemical evidence that this is truly the authentic cleavage site. Furthermore, your speculation about the evolution of the ability of the dengue protease to cleave STING is interesting, but the reviewers feel that alternative models need to be considered in the manuscript.Essential revisions:

Regarding the evolutionary model put forward, we have now highlighted the evolutionary model suggested by the reviewers, which we agree is more likely.

To address the location of the cleavage site, we have now immuno-precipitated the cleaved form of STING and performed mass spectrometry experiments. To do this experiment, we recruited two new authors who are proteomics experts (Webb and Old). Despite multiple attempts at this, the experiment did not allow us to precisely pin down the cleavage site. We were able to find some new data in the published literature that indicates the cleavage site is very close to the location that we have mapped (based on the molecular weights of the liberated products on a gel), and we discuss that.

As a result, we have tempered our language throughout the document, and now refer to STING positions 78/79 as a cleavage determinant, rather than a cleavage site. This seems like it should be a satisfactory solution to the cleavage site issue, since reviewer 1 stated that we should “provide biochemical evidence for cleavage at this site or alter their conclusions to state that they have identified sequences that are required, “and reviewer 2 stated “in the absence of direct evidence of the cleavage site, the authors should refer to this sequence as required for cleavage.” reviewer 3 did not comment on this issue.

With regards to one of the comments of reviewer #2, who wanted “proof that STING is cleaved in DENV infected cells” we have now performed a new experiment showing the cleavage of STING during infection, which is shown in in the new Figure 3—figure supplement 1.

Reviewer #1:This manuscript provides interesting results showing that the dengue virus infection can promote the cleavage of human STING and certain ape STING molecules but not many of the other primate STINGs. The research identified the human sequence needed for cleavage as 78RG79, which was also found in the ape proteins. Alteration of primate molecules to RG also allowed cleavage, and the authors conclude that this is the cleavage site. The results are very important in understanding the ability of dengue virus to infect humans and different non-human primate species.The conclusion that they have identified a new cleavage site seems to be premature in the absence of any biochemical evidence. The authors need to provide biochemical evidence for cleavage at this site or alter their conclusions to state that they have identified sequences that are required.

We agree. We now refer to this as a cleavage determinant rather than a cleavage site, and clearly state the limitations of our finding. Please see additional commentary on this point in the opening remarks to the editor above, describing efforts that we made to address this issue.

Second, they speculate in the abstract and elsewhere that "dengue viruses have evolved to fine-tune their pathogenicity in primates, but not yet in humans." I understand the evolution of a parasite to cause limited pathogenicity in its host to allow host survival, but where did the ability to cleave human STING come from? Was this from infection of an as yet unidentified species or did it evolve with infection of humans? This seems to be a very incomplete model.

We agree, and now highlight a better model. In the Discussion section we now state:

“Why do dengue viruses universally cleave human but not monkey STING? It’s possible that what we have uncovered is a brilliant method for balancing alternate host species, one of which is dense and abundant (humans), versus others that are spare and exist in smaller populations (primates in nature). In this scenario, dengue viruses have evolved to suppress innate immunity in humans in order to increase viral titers and spread, even though this trait comes at the cost of increased pathogenicity in some individuals. This might be a good strategy in our abundant and dense host population, where the fitness cost of severe disease in a fraction of individuals would be outweighed by excellent spread. Remarkably, though, dengue viruses have achieved this by evolved to recognize a portion of human STING that is not conserved in the STING of the wild and more rare animals that serve as their sustaining reservoir in nature, allowing the viruses to maintain decreased pathogenicity in these species. The evolution of the viral proteases to cleave human STING and simultaneously to avoid cleavage of monkey STING would be expected to reduce virus titers in monkeys, as the interferon pathway would be at least partially enabled. This may be beneficial for many reasons, one of which is that the production of a low-level innate immune response may allow the virus to replicate in reservoir host species without inducing high titers and strong adaptive immune responses.

Reviewer #2:This is an interesting manuscript that investigates in more detail the ability of the protease of DENV to cleave STING in a species-specific manner to evade induction of IFN responses. The authors find that most of the primates, including those that are a reservoir for DENV, have a STING unable to be cleaved by DENV. They found that this is due to a single amino acid change, most likely representing the cleavage site in STING of the dengue protease. In general, the data are consistent with this conclusion. However, some of the conclusions of the authors are too farfetched and they should be tempered.1) The authors are likely right in the cleavage sequence of STING, as one single amino acid change make human STING not cleaved and primate or mouse STING cleaved. However, there is always the possibility that this sequence participates in recognition by the protease and that cleavage occurs somewhere else. In the absence of direct evidence of the cleavage site, the authors should refer to this sequence as required for cleavage, but they cannot exclude that cleavage occurs at a different site.

We agree, and we now refer to this as a cleavage determinant rather than a cleavage site, and clearly state the limitations of our finding. Please see additional commentary on this point in the opening remarks to the editor above, describing efforts that we made to address this issue.

2) The experiments in Figure 3 clearly show that a single amino acid at STING determines the efficiency of inhibition of DENV by STING. Not shown in this figure is the number of DENV infected cells. This is important, as the authors did not see differences in levels of STING, which most likely indicate that the majority of the cells are not infected and therefore, that STING is not cleaved in most of the cells due to the lack of infection. In order to proof that STING is cleaved in DENV infected cells, the authors should sort DENV infected versus non-infected cells and determine the amount of STING in both populations as compared with a control host protein.

We have now performed an additional experiment and included a new supplemental figure (new Figure 3—figure supplement 1) showing the cleavage of STING in our cell lines during infection.

3) The speculation of the authors: "We propose that dengue viruses have evolved to fine-tune their pathogenicity in their monkey reservoir hosts, but not yet humans. Reduced viral pathogenicity within reservoir hosts is predicted to be beneficial to both host and virus" does not make sense. This will mean that the protease of DENV just by chance happened to cleave human STING, which also just by chance happened to be a major viral sensor. An alternative most likely explanation is that humans are since many thousands of years the main host for propagation of dengue virus, and therefore, the protease of DENV has evolved to cleave STING of the main host, humans, sacrificing the ability to cleave STING in primates, which are minor hosts, and therefore contribute less to the overall burden of dengue virus in nature. In other words, between two DENV, one cleaving human STING and the other cleaving primate STING, the one cleaving human STING has a competitive advantage over the one cleaving primate STING, as the virus has been mainly adapted to human and human mosquitoes, and primates might now be spillover reservoirs, but not main drivers of DENV evolution.

We now highlight this model in the paper.

Reviewer #3:

*It has been demonstrated that the DENV protease can cleave Human STING. Using an evolution-guided approach, the authors determine the ability of the DENV protease to cleave STING varies across species including primates due to sequence differences at amino acid positions 78 and 79. This finding is significant because it answers the long-standing question of why dengue does not replicate to high titers and display pathology similar to what is observed during human infection in primate models. I feel this is a thorough study, in particular, due to the use of genetic complementation, and the various STING orthologs and DENV (1-4) proteases assayed.*

Some modifications should be made to the text including the Title.

We have crafted a new title.